# Inverse mapping of quantum properties to structures for chemical space of small organic molecules

Alessio Fallani [1] ✉, Leonardo Medrano Sandonas [1,2] ✉ & Alexandre Tkatchenko [1] ✉

Computer-driven molecular design combines the principles of chemistry, physics, and artificial intelligence to identify chemical compounds with tailored properties. While quantum-mechanical (QM) methods, coupled with machine learning, already offer a direct mapping from 3D molecular structures to their properties, effective methodologies for the inverse mapping in chemical space remain elusive. We address this challenge by demonstrating the possibility of parametrizing a chemical space with a finite set of QM properties. Our proof-of-concept implementation achieves an approximate property-to-structure mapping, the QIM model (which stands for "Quantum Inverse Mapping"), by forcing a variational auto-encoder with a property encoder to obtain a common internal representation for both structures and properties. After validating this mapping for small drug-like molecules, we illustrate its capabilities with an explainability study as well as by the generation of de novo molecular structures with targeted properties and transition pathways between conformational isomers. Our findings thus provide a proof-of-principle demonstration aiming to enable the inverse property-to-structure design in diverse chemical spaces.

The discovery and optimization of chemical compounds can be accelerated thanks to the marked advancements in quantum and statistical methods, their implementation in advanced software, as well as the seemingly never-ending improvement in computer hardware[1,2]. Unlike the traditional painstaking trial-and-error process which heavily relied on experimental work (known as the Edisonian approach), we can now compute a wide range of accurate physicochemical properties of a given compound using quantum-mechanical (QM) methods with only the respective atomic coordinates and atom types. However, rationally exploring the incredibly vast chemical compound space (CCS, estimated to contain $10^{60}$ molecular structures even for small organic molecules)[3] via highly accurate QM methods is still unfeasible due to their prohibitive computational cost. In this regard, machine learning (ML) techniques have revolutionized the field of molecular

design by offering a fast but just as accurate method for obtaining properties from 3D molecular structures (a.k.a., direct mapping)[4–8], positioning them as an indispensable resource in high-throughput screening pipelines[9–15]. Although these approximate mappings have undoubtedly enhanced our understanding of the CCS, it is the possibility to invert them that has the potential to truly disrupt and transform the field. Addressing this challenge would allow us to predict the 3D molecular structures from their inherent properties, representing a paradigm shift in the design and discovery of chemical compounds with specific functionalities.

The quest to establish an inverse mapping has emerged as a formidable challenge, captivating the interest and dedication of researchers from various disciplines such as organic chemistry, materials science, and molecular docking[16–25]. A hint for the existence

[1]Department of Physics and Materials Science, University of Luxembourg, L-1511 Luxembourg City, Luxembourg. [2]Institute for Materials Science and Max Bergmann Center of Biomaterials, TU Dresden, 01062 Dresden, Germany. ✉e-mail: alessio.fallani.001@student.uni.lu; leonardoms20@gmail.com; alexandre.tkatchenko@uni.lu

**Fig. 1 | Scheme of our approach to inverse property–structure design. a** For training the QIM model (which stands for "Quantum Inverse Mapping"), we used a Variational Auto-Encoder (VAE) architecture which encodes each molecular structure **x** (atoms are represented as colored balls), in a distribution $q_\phi(\mathbf{z}|\mathbf{x})$ from which the latent $z$ is sampled and passed to a decoder to reconstruct the original structure. On the other hand, a property encoder is used to encode the associated quantum-mechanical (QM) properties **y** into a distribution $p_\psi(\mathbf{z}|\mathbf{y})$. These networks are jointly trained using the Evidence Lower Bound (ELBO) loss function that encompasses a molecular reconstruction term $-\log(p_\theta(\mathbf{x}|\mathbf{z}))$ and a Kullback- Leibler regularization term $D_{\mathrm{KL}}(q_\phi(\mathbf{z}|\mathbf{x})\|N(\mathbf{0}, \mathbf{1}))$, together with the additional loss component $-\log(p_\psi(\mathbf{z}|\mathbf{y}))$ for predicting the latent **z** from the properties **y**. The result of this joint training is a common latent representation for both properties and structures. **b** At inference time, one can combine the property encoder with the decoder component of the VAE and, hence, successfully approximate the Chemical Compound Space (CCS) parameterization using QM properties as intrinsic coordinates. The differentiability of our CCS parameterization enables us to identify the most relevant properties in the molecular reconstruction process as well as to perform a series of molecular design tasks.

of such a property-to-structure relationship is given by considering the demonstrated capability of generative models to selectively generate random structures conditioned on a predefined set of desired properties. Indeed, generative modeling has yielded numerous ground-breaking research outcomes[26–28], particularly in the field of cheminformatics, in which there is extensive literature delving into diverse generative architectures with research focusing mostly on language models and autoregressive generation employing a text-based representation like SMILES[29–40]. Models dealing with the intricate 3D structure of molecules have been recently developed, leading to promising results across different design tasks[33,41–47]. In the same breath, multi-objective optimization problems have been tackled using generative models and genetic algorithms, e.g., the design of functional organic molecules for optoelectronics applications and of candidate structures for dielectric organic materials[48–51]. These compelling examples underscore the relevance of tackling the inverse design problem, yet the potential of parametrizing the CCS using properties as coordinates remains unexplored. Overcoming this challenge would lay the groundwork for an alternative and multifaceted approach to understanding and manipulating the intricate relationship between the properties and structures of organic molecules.

Stemming from the "freedom of design" conjecture in the molecular property space espoused in our previous works[52,53], this study aims to investigate three main aspects of generative models for molecular design: (i) the definition of a differentiable parameterization of CCS by leveraging QM data of equilibrium molecules, (ii) the explainability of the resulting property-to-structure mapping, and (iii) applicability range of the learned inverse mapping. At present, not all

three of these points can be addressed using state-of-the-art models in conditional 3D molecular generation[42,43]. The primary reason for this lies in the inherent design of each model, which hinders the establishment of a differentiable property-to-structure mapping with respect to the molecular size, see "Methods" section for an extended explanation. Hence, we propose a more flexible and tailored approach, the QIM model (which stands for "Quantum Inverse Mapping"), that combines a Variational Auto-Encoder (VAE) architecture to encode the molecular structures (represented as Coulomb matrices) with a property encoder to encode the associated QM properties, see Fig. 1. The joint training of the VAE and property encoders yields a low-dimensional internal representation that is common for both the molecular structures and QM properties. This enables us to combine the property encoder with the decoder component of the VAE and, hence, to successfully approximate the CCS parameterization using QM properties as intrinsic coordinates for navigating the chemical space of small drug-like molecules contained in the QM7-X dataset[54]. The QIM model accurately predicts the heavy atom composition of molecules, reconstructs their geometries with reasonable accuracy, and meets the criteria of being one-shot and differentiable, making it explainable and flexible. Thanks to the differentiability of our CCS parameterization, we can identify the most relevant properties in the molecular reconstruction process as well as substructures in the molecular property space covered by QM7-X. The conventional multi-objective generative modeling paradigm is also retrieved by conditionally sampling in the input space of properties to accomplish two distinct design tasks with well-defined objectives, producing results comparable to a specialized model such as cG-Schnet[42] trained on the

same tasks. As a final showcase of the capabilities of learning a CCS parameterization, we have implemented a geodesic search algorithm that uses the latent space representation as internal coordinates, enabling the definition of transition structures and the estimation of rotational energy profiles by learning only from equilibrium geome-

tries. Although our proof-of-concept implementation is currently assessed only for small molecules, our findings suggest that a CCS parameterization based on QM properties is feasible and holds promise for a wide range of applications, including interpretability, generation of de novo molecules, transition path interpolation, and reaction barrier estimation. Thus, this work highlights the remarkable opportunities that arise from defining an inverse mapping approach to advance our understanding of the chemical compound space by connecting QM properties and molecular structures.

## Table 1 | Quantum mechanical (QM) properties

| Symbol | Property description | Units | Type | Class |
|---|---|---|---|---|
| $E_{AT}$ | Atomization energy | eV | M,G | E |
| $E_{MBD}$ | MBD energy | eV | M,G | E |
| $E_{XX}$ | Exchange energy | eV | M,G | E |
| $E_{NN}$ | Nuclear-nuclear energy | eV | M,G | E |
| $E_{EE}$ | Electron-electron energy | eV | M,G | E |
| $E_{KIN}$ | Kinetic energy | eV | M,G | E |
| $E_{GAP}$ | HOMO-LUMO gap | eV | M,G | I |
| $E_{HOMO}^{0}$ | HOMO energy | eV | M,G | I |
| $E_{LUMO}^{0}$ | LUMO energy | eV | M,G | I |
| $E_{HOMO}^{1}$ | HOMO-1 energy | eV | M,G | I |
| $E_{LUMO}^{1}$ | LUMO+1 energy | eV | M,G | I |
| $E_{HOMO}^{2}$ | HOMO-2 energy | eV | M,G | I |
| $E_{LUMO}^{2}$ | LUMO+2 energy | eV | M,G | I |
| $\zeta$ | Total dipole moment | $e \cdot Å$ | M,G | I |
| $\alpha$ | Isotropic molecular polarizability | $a_0^3$ | M,R | E |
| $D_{MAX}$ | Maximum atom-atom distance | Å | S,G | I |

List of QM properties (and corresponding symbols) taken from the QM7-X dataset[54] and considered during the training of our model, see Fig. 1. In the units provided for each of these QM properties, $a_0$ represents the atomic units of length (Bohr radius). All properties are scalars, i.e., their dimension is 1. Property types and classes were categorized as follows: structural (S), global/molecular (M), ground-state (G), response (R), extensive (E), and intensive (I).

## Results
### Property-to-structure mapping
To train the VAE and the property encoder (see "Methods" section), we have considered a subset from QM7-X dataset of 40,988 equilibrium conformations of molecules with up to seven heavy atoms including C, N, and O. Here, 17 QM global extensive and intensive properties (listed in Table 1) per conformation were selected to define the property encoder. The technical details of the input data and training procedure are specified in the "Methods" section and Supplementary Note 3 of the Supplementary Information (SI), respectively.

We now assess the capability of the trained QIM model to establish an approximate parameterization of the chemical compound space (CCS) spanned by QM7-X based on the predefined set of QM properties. In doing so, we will make use of the molecules in the test set together with their corresponding properties, i.e., the property set of these molecules will be used to construct the model and, then, the generated molecule will be compared with the original one. Figure 2a shows the boxplots of the relative error on Coulomb matrix (CM) reconstruction for an increasing number of properties used to train the model. Here we have analyzed the relative error on CM instead of the root-mean-squared deviation (RMSD) between structures because the latter is only defined for molecules for which the composition is correctly predicted and, consequently, it will present more fluctuations across different number of properties especially when the number of properties used is low and hence the errors in composition are high. Accordingly, one can see that our model allows us to

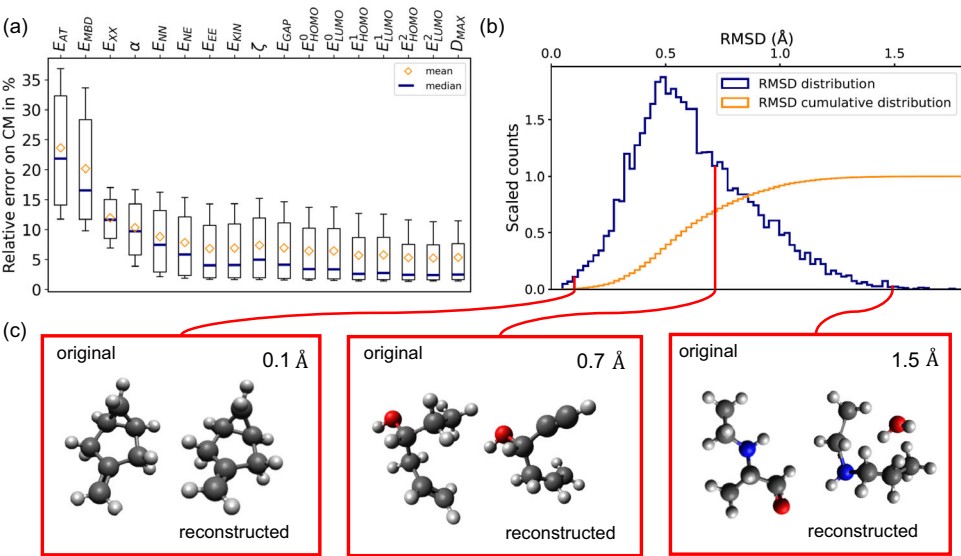

**Fig. 2 | Performance of the QIM model in the molecular reconstruction process. a** Boxplots of the relative error on the reconstruction of Coulomb matrix (CM) representation of structures from QM properties as a function of the number of molecular properties considered during the training. See Table 1 for the definition of the property symbols. The considered data per box is the full test set ($n = 10,988$ molecules). For better visibility, we consider the whiskers to extend from the 15th to the 85th percentile. **b** Frequency plot of the root-mean-squared error (RMSD) between the geometries retrieved from the reconstructed CMs and corresponding

original representations. The cumulative function of this plot is also presented, see orange curve. **c** Selected examples for different RMSD values to compare the original molecule with this one reconstructed from a predefined set of QM properties. We empirically found that RMSD < 0.7 Å; is an adequate threshold to separate molecules with an acceptable reconstruction of the heavy atom structure in terms of topology and orientation. The chosen colors to represent the atoms are gray for carbon, red for oxygen, blue for nitrogen, and white for hydrogen. Source data are provided as a Source Data file.

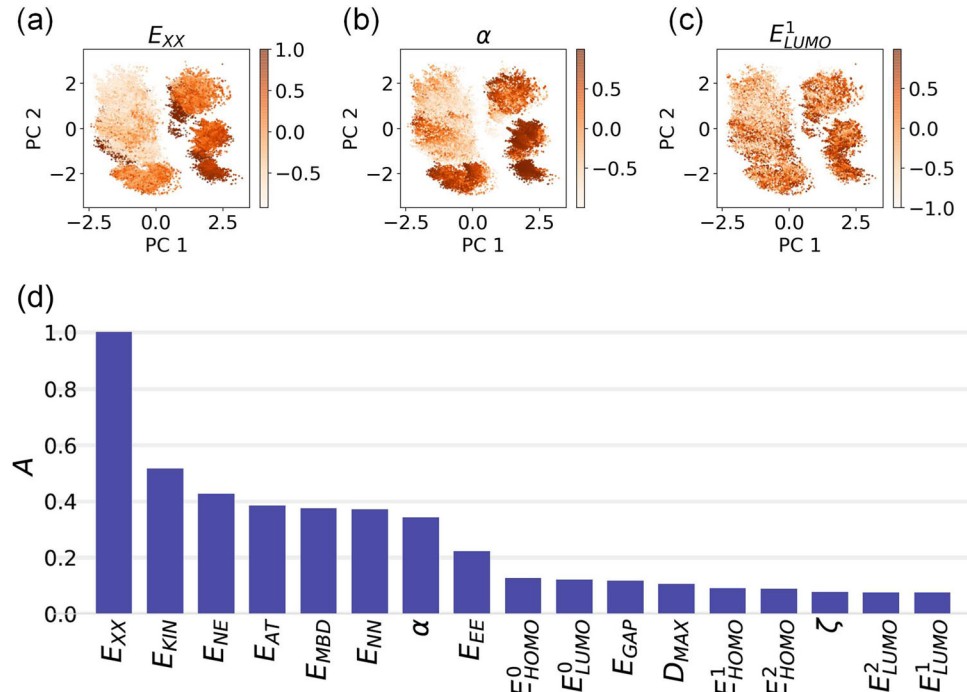

**Fig. 3 | Gradient attribution map for molecular properties.** Two-dimensional principal component analysis (PCA) of the latent space of the VAE (which stands for Variational auto-encoder) encoder, colored respectively with the values of (**a**) $E_{XX}$, (**b**) $\alpha$, and (**c**) $E^1_{LUMO}$. For the color scheme, we rescaled the property values and applied a hyperbolic tangent function to avoid obscuration effects from outliers. **d** The contribution of each property to the output structure is evaluated by defining the variable $A$, which considers the partial derivative of the reconstructed Coulomb Matrix (CM) with respect to a given property (see Eq. (4)). Here, one can see that extensive properties are more relevant than intensive ones for molecular reconstruction. See Table 1 for the definition of the property symbols. Source data are provided as a Source Data file.

reconstruct the representation with an average error that converges to ~5% when considering more than seven properties during the training, after which the distribution becomes increasingly skewed as can be seen by the median value. This relative error is defined as $\Delta = \frac{|\tilde{\mathbf{C}} - \mathbf{C}|}{|\mathbf{C}|} \times 100$, where $\mathbf{C}$ is the original CM and $\tilde{\mathbf{C}}$ is the reconstructed one (both to be considered vectors). While we provide the mean and deviation of the CM reconstruction error, it is important to note that this indicator does not completely reflect the quality of the mapping due to its noisy and nonlinear correlation with RMSD (see Supplementary Fig. 1 of SI).

To have a better understanding of this finding, we also report the distribution and the cumulative distribution over the test set for the RMSD in the case of the full set of properties (see Fig. 2b). The performance of the QIM model deteriorates when considering a lower number of properties, as it is shown in Supplementary Fig. 2 of the SI. Indeed, the mode of the RMSD distribution, when considering the full set of properties, is close to 0.5 Å. Overall, despite obtaining a wide RMSD spectrum ([0.05,1.6] Å), we found that ~70% of the molecules in the test set were reconstructed within RMSD = 0.7 Å. We empirically found this threshold to be adequate to separate molecules with an acceptable reconstruction of the heavy atom structure in terms of topology and orientation. To further motivate the selection of this threshold, we provide several illustrative examples of both original and reconstructed molecules in Fig. 2c, along with a quantitative analysis of topological errors in Supplementary Fig. 3 of the SI. Concerning the chemical composition reconstruction, the model exhibits excellent performance, correctly predicting it for 99.96% of the molecules in the test set. Additionally, we investigated the impact on the performance of the QIM model when considering extensive and intensive properties separately in the training procedure, see Supplementary Fig. 4 of the SI. The models trained on separated properties present a higher number of structures with large RMSD compared to the final model,

indicating a lower performance in structure reconstruction. In fact, the model trained on intensive properties was only able to reconstruct with the right chemical composition approximately 8000 from 10,000 molecules of the test set. This decrease from 99.96 to 75.85% in molecular reconstruction comes together with an increment of the ⟨RMSD⟩ up to 0.7 Å as well as a reduction up to 55% of tested molecules reconstructed with an RMSD below 0.7 Å. In brief, these results have confirmed the need to use both types of properties in the training of the QIM model to have a better reconstruction of molecular structures.

## Interpretability and performance of our model

Then, the established CCS parameterization is analyzed by implementing a gradient attribution map, which enables us to assess the individual contributions of each property to the output structures. The calculation of an attribution map $A$ for each property is explained in "Methods" section. Figure 3 shows the $A$ values per property which have been normalized over the maximum value and sorted in decreasing order for the full representation. Overall, we have found that extensive properties are more informative than intensive ones for the task of molecular reconstruction. This can be explained by the fact that these extensive properties depend on crucial molecular features that are also considered in a 3D representation like CM, e.g., number of atoms, number of electrons (related to the chemical composition), and geometry. Moreover, when comparing CMs, even a slight difference of one atom can significantly increase the loss, leading to a larger sensitivity of the model to variations in system size and composition. Thus, $A$ values for the components of the total energy and the molecular polarizability are higher compared with these for the molecular orbital energies and dipole moment; in particular, $E_{XX}$ and $E_{KIN}$ present the largest $A$ values. Interestingly, this finding is in agreement with the identification of molecular clusters in the two-dimensional principal component analysis (PCA) of the latent space of the VAE encoder, i.e.,

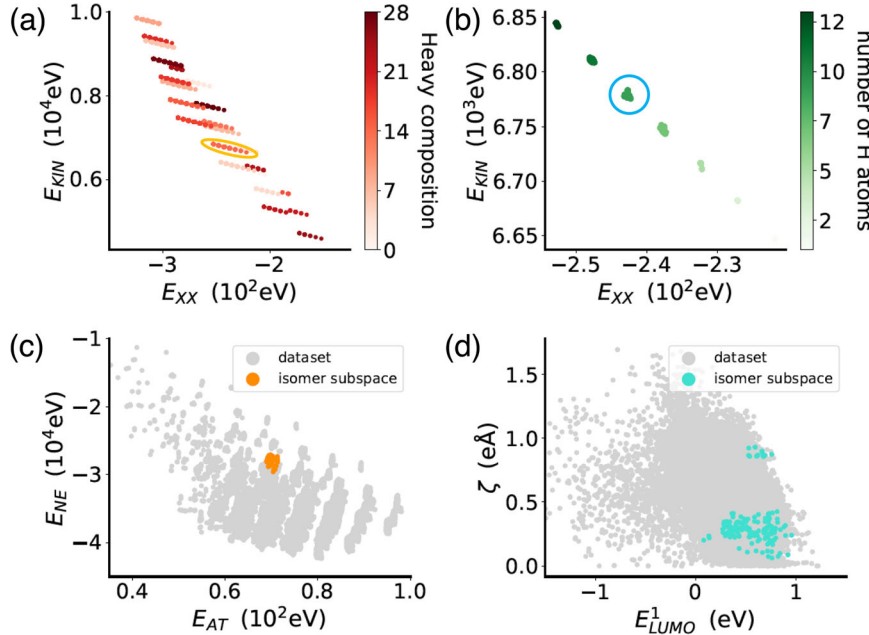

**Fig. 4 | Two-dimensional (2D) projections of the high-dimensional QM7-X molecular property space.** We have studied how the revealed hierarchy of QM properties (see Fig. 3) organizes the QM7-X chemical compound space. **a** 2D property space defined by the two properties with highest $A$ values, i.e., $(E_{XX}, E_{KIN})$-space. Here, the molecules in the dataset seem to organize as linear-shape clusters containing molecules with the same heavy atom composition. The color code used here is obtained by assigning an integer number to each heavy atom composition present in the dataset. **b** A closer examination of one of the highly populated clusters in the $(E_{XX}, E_{KIN})$-space (see yellow ellipse in panel (**a**)).

Molecules are highlighted with respect to their number of H atoms. We then picked a molecular isomer subspace (see blue ellipse in (**b**)) and plotted how different pairs of QM properties with (**c**) high and (**d**) low $A$ values distribute the molecular structures. This analysis demonstrated that high $A$ values identify properties that are better local coordinates for exploring this subspace, i.e., these properties can be used to distinguish molecular structures within a specific molecular isomer sub-space. See Table 1 for the definition of the property symbols. Source data are provided as a Source Data file.

the higher the $A$ value, the more correlated the property with respect to the PCA (see insets in Fig. 3). A similar observation holds when using a more complex dimensionality reduction technique such as t-distributed stochastic neighbor embedding (t-SNE)[55], see Supplementary Fig. 6 of the SI. The fact that a linear method such as PCA shows good organization in terms of energy-related properties is a non-trivial finding that will be later exploited in the interpolation of transition structures.

Furthermore, we have studied how the revealed hierarchy of QM properties organizes the QM7-X CCS. Starting from the properties with the highest $A$ value, in Fig. 4a, one can see the two-dimensional projection of the QM7-X molecular property space defined by $E_{KIN}$ and $E_{XX}$, i.e., $(E_{KIN}, E_{XX})$-space. Despite these two properties having a high inverse correlation (Pearson coefficient = − 0.92), it is noticeable how the molecules in the dataset seem to organize as linear-shape clusters containing molecules with the same heavy atom composition. In particular, upon examining those clusters, it becomes evident that $E_{KIN}$ is mostly influenced by the heavy atom composition within a molecule. On the other hand, $E_{XX}$ is highly sensitive to the number of H atoms, thereby indicating a dependence on the particular bond types that are present. This is further analyzed in Fig. 4b, where we provide a closer examination of one of the highly populated clusters (see yellow ellipse in plot) and highlight the molecules with respect to their number of H atoms. Here, we uncovered a finer local structure with almost perfect inverse correlation (Pearson coefficient = − 0.99) as well as very compact clusters formed by isomers. Overall, this behavior can be understood by considering the qualitative aspects of $E_{KIN}$ and $E_{XX}$: the dominant contribution to $E_{KIN}$ stems from the inner shell electrons (trivial consequence of the virial theorem) and the primary influence on $E_{XX}$ arises from the valence electrons. Also, exchange-related quantities have been found to play a significant role in characterizing bonds[56,57], explaining their sensitivity to the number of H atoms in a

molecule. These considerations account for the efficacy of these two QM properties in identifying clustered molecular isomer subspaces and accurately predicting heavy atom composition.

Next, we pick a molecular isomer subspace (see blue circle in plot) and show how different pairs of QM properties with high and low $A$ values distribute the molecular structures (see Fig. 4c, d). As expected, high $A$ values identify properties that are better local coordinates for exploring this subspace (*vide supra*). Indeed, these properties present relatively smaller changes in their values across related structures in comparison to properties with low $A$ values—an example of their efficiency in identifying molecular structures within a specific molecular isomer subspace while effectively distinguishing them from other structures spanning the entire property spectrum. Notice that the same behaviors can be found throughout the entire set of QM7-X equilibrium molecules. These findings already provide compelling evidence for the potential of our proof-of-concept implementation in furthering our understanding of the molecular property space and unraveling the intricate relationship between QM properties and molecular structures.

The successful property-based parameterization of CCS achieved by the model is grounded in the remarkable similarity attained among the latent representations of CMs and respective properties. A visualization that demonstrates this observation can be seen in Fig. 5a, which showcases the overlap between the PCA of both latent representations. Similar result was obtained by using the t-SNE technique, see Supplementary Fig. 7 of the SI. This verifies the initial assumption we stated about the joint training procedure, see "Methods" section. Moreover, we have examined how the differences in latent space representation correlate with the quality of the reconstructed molecules. This analysis aims at obtaining a self-consistent method for error estimation. To this end, if **z** is the latent representation from the property encoder, we can take the reconstructed representations and

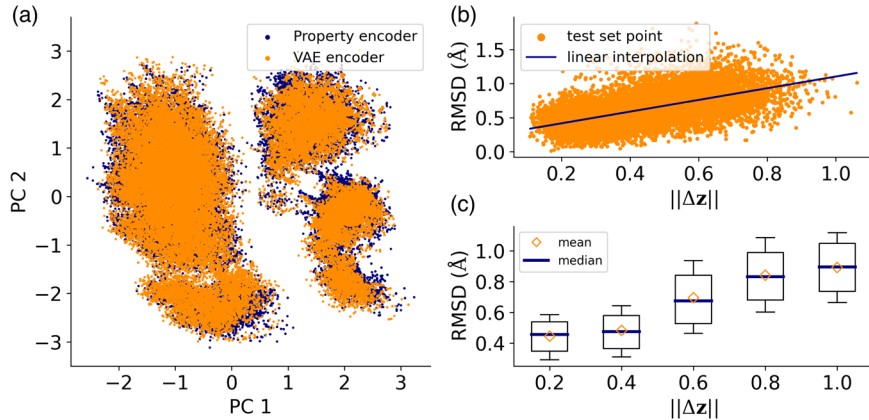

**Fig. 5 | Similarity analysis between the latent space representation from the VAE and property encoders. a** Overlap between the two-dimensional principal component analysis (PCA) of both latent space representations. **b** Correlation plot of the root-mean-squared error (RMSD) between the original and the reconstructed structures versus the latent space difference $||\Delta \mathbf{z}|| = ||\mathbf{z} - \tilde{\mathbf{z}}||$ for molecules in the test set. Here, $\mathbf{z}$ is the latent representation from the property encoder, while $\tilde{\mathbf{z}}$ is a new latent representation produced by taking the reconstructed representations and encoding them again with the VAE encoder. **c** Boxplots of the RMSD corresponding to different values of latent space difference $||\Delta \mathbf{z}||$. The data points considered per boxplot are all the ones in the test set with RMSD in an interval of ± 0.1 Å of the indicated label. For better visibility, we consider the whiskers to extend from the 15th to the 85th percentile. Subsequently, since we are looking for high-quality generated molecules that have low RMSD, we define the interval $||\Delta \mathbf{z}|| \in [0,0.4]$ to filter out molecules during the molecular generation process. Source data are provided as a Source Data file.

encode them again with the VAE encoder, producing a new latent representation $\tilde{\mathbf{z}}$. Then, we look at the correlation of the quantity $||\Delta \mathbf{z}|| = ||\mathbf{z} - \tilde{\mathbf{z}}||$ with the RMSD between the original structures and the one reconstructed from properties considering the molecules in test set. In Fig. 5(b), one can see that there seem to be a bulk (approximately linear) correlation with the presence of numerous outliers. To further investigate this behavior, we also look at the boxplots of the RMSD for varying values of $\Delta \mathbf{z}$ by splitting the test set into subsets to get an adequate statistics for each point. The results are plotted in Fig. 5(c) in which we show that there is a nonlinear but monotonic behavior for the relationships between these quantities, finding a minimum for RMSD in the region of low $||\Delta \mathbf{z}||$ values ($\in [0,0.4]$). Since for reconstructing the actual molecular structure, we are mostly interested in having a low RMSD, we will filter out the generated structures for which $||\Delta \mathbf{z}||$ falls outside this predetermined interval in the following sections. This screening approach based on error estimation will be employed in the next application tests to enhance the quality of generated structures with a targeted set of QM properties.

**Acting as a conditional generative model**

The learned CCS parameterization based on QM properties can provide a versatile solution for multi-objective targeted molecular generation. Here, we show the capability of the QIM model to target predefined pairs of QM properties by using a multivariate Gaussian regression approach to model the distribution of the QM7-X property space (see "Methods" section). Based on this procedure, we first navigate through the $(\alpha, E_{\text{MBD}})$-space that encompasses two relevant properties for molecular reconstruction (high attribution value $A$), see Fig. 6a. The moderate degree of correlation between these properties of QM7-X molecules (i.e., Pearson coefficient = 0.60, gray dots) also grants us access to the intrinsic "freedom of design" in CCS when searching for molecules with desired functionality[52]. Subsequently, we transition to a weakly correlated 2D property space given by $(\alpha, \zeta)$-space (i.e., Pearson coefficient = 0.44, gray dots) by replacing $E_{\text{MBD}}$ with a dipole moment $\zeta$ (low $A$), see Fig. 6b. In the $(E_{\text{EMBD}}, \alpha)$ property space, fifteen samples (or 15-molecule set) were generated per targeted pair, while ten samples (or 10-molecule set) were generated per target in the $(\zeta, \alpha)$ one. The targets are represented as colored crosses with capital letters. The targeted $\alpha$ values were here selected to

generate medium-to large-sized molecules, since $\alpha$ is an extensive property that mostly depends on the elemental composition as well as the connectivity of atoms within the molecule. Whereas, the targeted $E_{\text{MBD}}$ and $\zeta$ values were chosen to tune the topological effects (i.e., extended vs. compact, packed/globular vs. void space) and elemental composition (homogeneous vs. heterogeneous) in the generated molecules, respectively. The top 5 generated molecules (i.e., five optimal molecules) per target have been highlighted with colored circles in both 2D property spaces. In Fig. 6, we also show the generated structure of a selected molecule per target. Overall, these results demonstrate that our model is capable of generating diverse molecular structures with similar scaffolds as in QM7-X molecules by using only QM properties as input.

Given the discrete nature of CCS and particularly the reduced coverage of the employed QM7-X dataset, it is clear that the inverse mapping generates molecules that can deviate from the respective targeted values. The performance per target can be quantitatively measured by defining $\epsilon = \frac{|y_{\text{calc}} - y_{\text{t}}|}{\Delta y} \times 100$ with $y_{\text{calc}}$ and $y_{\text{t}}$ as the calculated and target values of the property $y$. $\Delta y$ represents the extent of the property spectrum across the entire dataset. Accordingly, for each of the 15-molecule set depicted in Fig. 6a, the $\epsilon$ value is circa 9.2% for $E_{\text{MBD}}$ and circa 3.5% for $\alpha$. These values are reduced to circa 3.2% for $E_{\text{MBD}}$ and circa 1.3% for $\alpha$ by only considering the top 5-molecule set. On the other hand, the molecules generated in $(\alpha, \zeta)$-space displayed $\epsilon \approx 2.8\%$ for the prediction of $\alpha$ but the $\epsilon$ corresponding to $\zeta$ was circa 5.9% (see top 5-molecule set per target in Fig. 6b). Despite preventing the establishment of a differentiable property-to-structure mapping with respect to molecular size, we have used the state-of-the-art architecture cG-SchNet[42] to rigorously examine our results in the two multi-property targeted molecular design tasks. Although cG-SchNet models achieve better performances, the QIM model remains comparable, as the difference in the values of $\epsilon$ is not substantial, ranging from 0.6 to 5.3% in favor of cg-Schnet (see Supplementary Note 5 and Supplementary Tables 1–4 of the SI). The only exception was the prediction of $E_{\text{MBD}}$ for the top 15 molecules in the $(\alpha, E_{\text{MBD}})$-space where the difference of $\epsilon$ values was $\approx 7.3\%$. This result can be related to the explicit treatment of H atoms during the training of the cG-SchNet model, while in our model H atoms are added after the molecule is generated (see "Methods" section). Notice that, in terms of computational efficiency, our model achieves convergence in three hours,

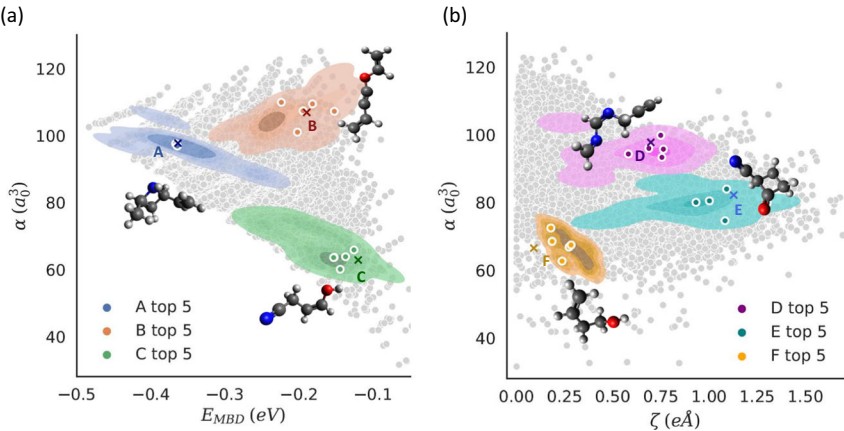

**Fig. 6 | Generation of molecules with a targeted pair of quantum-mechanical (QM) properties.** Here, we show the distribution of the generated molecules per targeted pair of QM properties. The targets A, B, C, D, E, and F are identified by colored crosses, and the distribution of the generated samples per target is represented as a density fit, with a darker shade indicating higher sample density. This is done in the 2D property space defined by (**a**) molecular polarizability and MBD energy (i.e., $(\alpha, E_{MBD})$-space) and (**b**) molecular polarizability and total dipole moment (i.e., $(\alpha, \zeta)$-space). To reconstruct these molecules from their QM properties, we have followed the molecular generation procedure explained in "Methods" section. The top 5-molecule set (i.e., five optimal molecules) per target have been highlighted with colored circles in each panel. For reference, the values

corresponding to QM7-X molecules are shown in the background (gray dots). Panels (**a**) and (**b**) contain a single structure per target that represents the top 5-molecule set (other structures are depicted in Supplementary Fig. 8 of the SI). These results demonstrate that the QIM model (which stands for "Quantum Inverse Mapping") displays a better performance for molecular reconstruction when only targeting extensive properties (see the spread of colored circles with respect to targeted values), which can be primarily attributed to the purely geometrical/chemical definition of the Coulomb matrix representation. The chosen colors to represent the atoms are gray for carbon, red for oxygen, blue for nitrogen, and white for hydrogen. Source data are provided as a Source Data file.

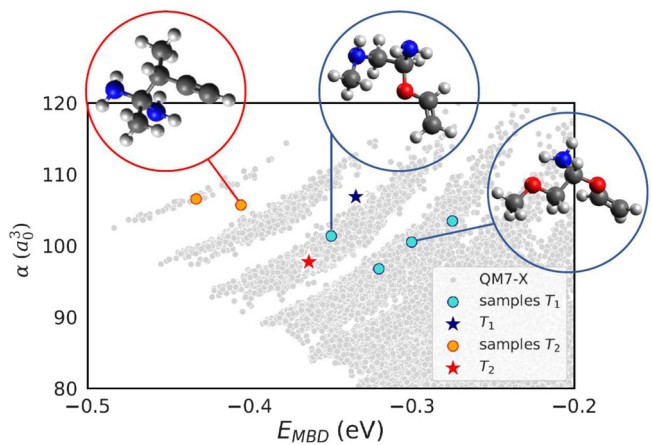

**Fig. 7 | Targeting unseen compounds with a given pair of molecular properties.** Evaluation of the impact of the modified molecular generation procedure on multiple targets across the 2D property space defined by molecular polarizability and MBD energy, i.e., $(\alpha, E_{MBD})$-space. To do so, we have retrained the QIM model (which stands for "Quantum Inverse Mapping") with a bias towards treating molecular fragments more independently (see more details in "Methods" section). In the graph, we plot the targets $T_1$ (blue star) and $T_2$ (red star) and their associated molecules with eight heavy atom compositions that were generated using the new model (colored circles). For reference, the values corresponding to QM7-X molecules are shown in the background (gray dots). Also depicted are selected novel molecular structures per target. The chosen colors to represent the atoms are gray for carbon, red for oxygen, blue for nitrogen, and white for hydrogen. Source data are provided as a Source Data file.

which is faster than the time required to train the cG-SchNet model on the same hardware (i.e., four days).

Furthermore, we have found that the spread of the generated molecules (i.e., flexibility of the model in molecular generation) can be rationalized by analyzing the relative variance of the conditional multi-Gaussian distributions of the non-targeted QM properties. In fact, molecules generated with the highest precision (i.e., targets A and D)

are characterized by lower negative log likelihood values and small relative variances in the extensive non-targeted properties (see Supplementary Fig. 4 of the SI). This outcome could be a key factor for controlling the degree of flexibility when designing molecules in targeted regions of a given property space since a larger variance in extensive properties may result in a more diverse set of molecules as these properties are the most relevant in defining the molecular structure (see Fig. 3). In this regard, taking a closer look at the generated molecules, one can see that the molecules in the $(E_{MBD}, \alpha)$-space display a greater diversity in heavy atom composition compared to the ones generated in the $(\zeta, \alpha)$-space, see Supplementary Fig. 8 of the SI. Importantly, the presence of diverse chemical compositions and structures within the larger sets for each target is another evidence that our approach does not follow a conventional chemical exploration based on fixed molecular scaffolds. Instead, our inverse design procedure can fully utilize the diversity of property-to-structure relations as embodied in the recently proposed "freedom of design" conjecture in CCS[52].

## Generating compounds in uncharted CCS regions

To demonstrate that the QIM model is capable of generating de novo molecules with desired QM properties beyond the QM7-X dataset, we have modified the training procedure to reduce the bias of the model towards treating existing molecular fragments (see "Methods" section). This action slightly reduces the model performance in molecular reconstruction (⟨RMSD⟩ is only decreased by 0.05 Å), but it enables the generation of scaffolds beyond QM7-X, mostly composed of unseen molecules containing eight heavy atoms. While using this new procedure across multiple targets in the $(\alpha, E_{MBD})$-space, we found that the generation of unseen compositions is confined to a region of low coverage defined by high $\alpha$ and large $|E_{MBD}|$ (see Fig. 7). Certainly, the model here exhibits a higher degree of flexibility and it is capable of generating molecules with diverse compositions of eight heavy atoms. In Fig. 7, one can also see that the spread of generated molecules for target $T_1$ and $T_2$ is broader than that of corresponding targets discussed in the preceding section. In term of the relative error $\epsilon$, the samples in target $T_1$ presented an error circa 10% for $E_{MBD}$ and 8% for $\alpha$,

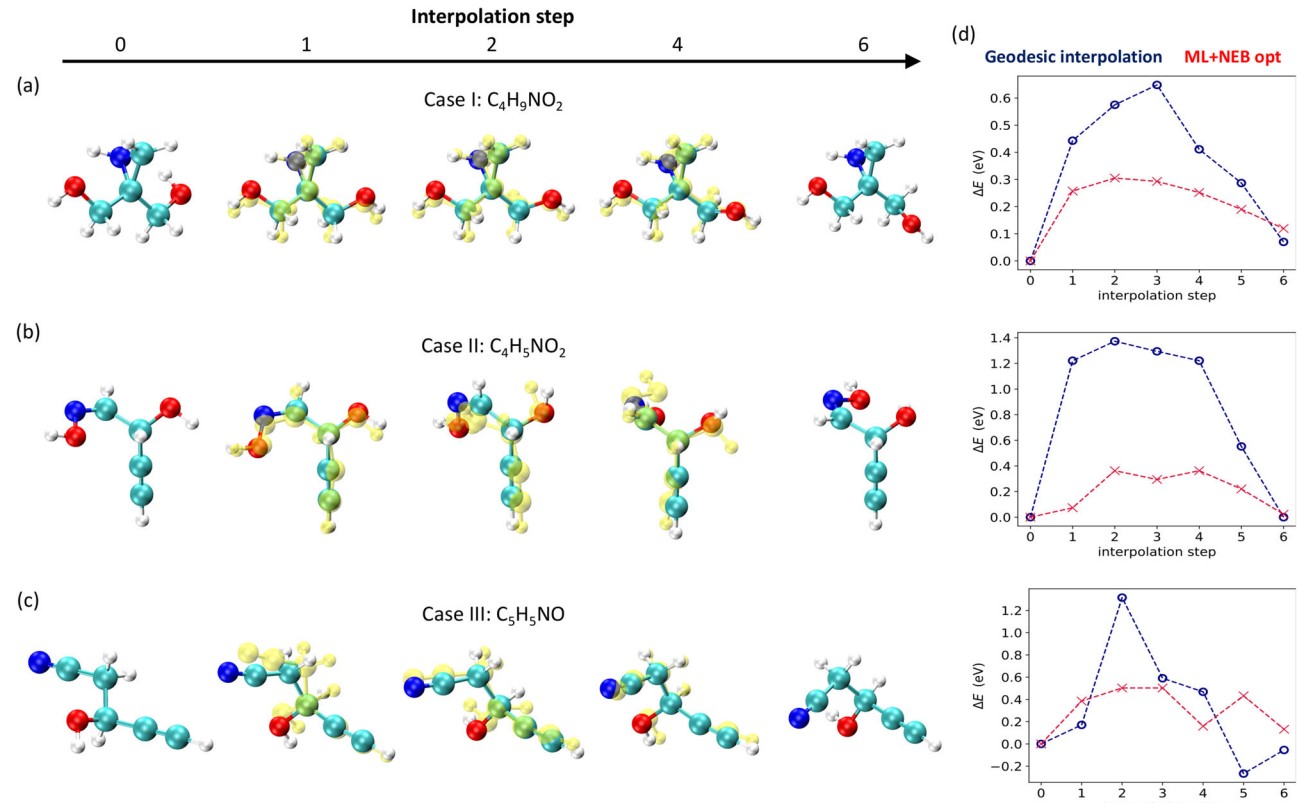

**Fig. 8 | Predicting transition path between conformational isomers.** By using the geodesic interpolation algorithm for VAEs (which is short for Variational Auto-encoders), we were able to generate interpolated geometries for three different pairs of conformational isomers (**a**) $C_4H_9NO_2$ (case I), (**b**) $C_4H_5NO_2$ (case II), and (**c**) $C_5H_5NO$ (case III). The molecular structures generated by the QIM model (which stands for "Quantum Inverse Mapping") are represented by yellow balls. For comparison, we show the new transition structures (solid colored balls) obtained by running machine learning-based nudged elastic band (ML-NEB) calculations using as initial guesses the previous interpolated geometries. The chosen colors to represent the atoms are cyan for carbon, red for oxygen, blue for nitrogen, and white for hydrogen. **d** Variation of the relative energy component $\Delta E_i = E_i - E_0$ ($E$ is the sum of atomization and many-body dispersion energies) as a function of the interpolation step $i$ for the three isomerizations shown in (**a**–**c**). We here show the results obtained for the corresponding geodesic in property space (blue curve) and the updated transition structures computed using the ML-NEB method (red curve). These findings are compelling evidence that latent representation can be used as an intrinsic coordinate system to generate reasonable guesses for the transition geometries between conformational isomers in QM7-X. Source data are provided as a Source Data file.

while, for samples in $T_2$, the observed errors were circa 20% for $E_{MBD}$ and 16% for $\alpha$. Despite the lower prediction accuracy of QM properties, the errors for target $T_1$ are notably comparable to the results shown in Fig. 6 and those obtained by the cG-SchNet model (see Supplementary Note 5 of the SI). Note that, unlike the QIM model, the cG-SchNet model can generate molecules with up to nine heavy atoms without requiring any additional modifications. Moreover, by analyzing the generated molecules using the QIM model, we found that the novelty in heavy atom compositions can vary depending on the specific target location, e.g., $T_2$ samples consider only (C,N)-based molecules while $T_1$ samples are more chemically diverse and cover (C,N,O)-based molecules. For both sets of samples, $E_{MBD}$ and $\alpha$ values are in the range of expected values for molecules with eight heavy atoms, i.e., $-0.6$ eV $< E_{MBD} < -0.09$ eV and 66 a.u. $< \alpha < 160$ a.u. Likewise, all generated molecules were well-aligned with chemical and physical intuition, e.g., molecules generated in target $T_1$ exhibit a larger $|E_{MBD}|$ than those in target B (see Fig. 6a) due to the more complex heavy atom composition and compact structure.

**Interpolating transition geometries between isomers**
Throughout the preceding sections, we have showcased the versatility of an approximate CCS parameterization in a range of contexts. Indeed, it was demonstrated that the latent space of the QIM model exhibits a discernible structure characterized by energetic properties with high $A$ values when undergoing a linear transformation such as

PCA, see Fig. 3. This particular behavior assures that a linear interpolation in latent space will generate structures for which the energetic properties smoothly change—a consequence of the fact that linear transformations do not affect convexity properties up to a sign. Accordingly, we here aim to explore the potential of utilizing this latent representation as an intrinsic coordinate system to generate reasonable guesses for the transition geometries between conformational isomers in QM7-X. To do so, we use the geodesic interpolation algorithm[58] for VAEs but, in this instance, the objective is to find curves in the property space that are geodesics with respect to the metric induced by the latent space encoding, see details of implementation in the "Methods" section. Following the selection criteria described in "Methods" section, we have investigated three different pairs of conformational isomers $C_4H_9NO_2$ (case I), $C_4H_5NO_2$ (case II), and $C_5H_5NO$ (case III) whose structures are achiral and were reconstructed with an RMSD $\leq 0.2$ Å, see initial ($i = 0$) and final ($i = 6$) geometries per case in Fig. 8a–c.

Overall, the interpolated geometries displayed in Fig. 8a–c (see yellow balls) effectively demonstrate the capability of the QIM model to produce plausible geometries for the transition path of the studied isomerizations. However, by analyzing sample by sample, we detected a few abrupt/unphysical changes in the geometries for the cases II and III between steps 1 and 2. These artifacts primarily arise from two key factors: (i) the sensitivity of the CM representation to small changes which could produce large mirror-like transformations in the resulting

geometry and (ii) the fact that model performance is degraded across unknown sectors of the latent space associated to the transition geometries. The relative energy component $\Delta E_i = E_i - E_0$ ($E$ is the sum of atomization and MBD dispersion energies) of the corresponding geodesic in property space for the three isomerizations is reported in Fig. 8d. Unexpectedly, with no request other than minimizing the distance in latent space, a barrier-like behavior is retrieved in all cases with energy barriers between 0.6 eV and 1.4 eV—another evidence of the potential application of our model for generating guesses of transition geometries as well as energy profiles. The obtained results are remarkable since the model was trained exclusively on equilibrium geometries.

To examine how close the estimated energy profile is to the true minimum energy path and the quality of the generated transition geometries, we have used them as initial guesses for a nudged elastic band (NEB) calculation following the ODE method[59]. Before proceeding with the NEB calculations, H atoms were added to all molecules with OpenBabel and, subsequently, optimizations of initial and final geometries were carried out employing an ML force field trained on PBE0+MBD energies/forces of equilibrium and non-equilibrium molecular conformations contained in QM7-X dataset (model taken from ref. 60). This accurate ML force field was also used to perform the NEB calculations at PBE0+MBD level of theory[61]. The relative energies calculated following this direct procedure are presented in Fig. 8d and compared to their corresponding geodesic energies, showing how the later were consistently overestimating the energy barrier. As for the updated geometries, the RMSD of their heavy atoms structures with respect to the initially interpolated ones was found to be between 0.14 Å and 0.35 Å, see colored balls in Fig. 8a–c.

## Discussion

In the present work, we have presented a machine-learning approach, the QIM model, to the inverse property-to-structure design process by learning a parameterization of the chemical compound space (CCS) of small organic molecules that uses QM properties as intrinsic coordinates to navigate the space of molecular structures. This challenging task was successfully accomplished through the development of a proof-of-concept implementation that jointly trains a variational autoencoder and a property encoder. The trained model, utilizing the equilibrium molecules contained in QM7-X dataset, is able to reconstruct the molecules (heavy atom composition and 3D structure) within the test set from their properties with reasonably good accuracy. Moreover, the differentiability of the learned CCS parameterization not only enabled us to identify the most relevant properties in the molecular reconstruction process but also revealed intriguing insights. Our findings indicate that the combination of exchange energy and kinetic energy plays a pivotal role in clustering the molecules in the dataset based on chemical composition and bond types. This information can be further utilized to conditionally generate molecules belonging to specific isomer subspaces within property spaces defined by more pertinent QM properties. In contrast, we observed that intensive properties exhibit limitations as local coordinates for conformational space navigation when compared to extensive ones—a result primarily attributed to the purely geometrical/chemical definition of the Coulomb Matrix (CM) representation. Nevertheless, the QIM model exhibited its best performance when both extensive and intensive properties were considered in the training procedure, highlighting the significance of intensive properties in achieving efficient molecular reconstruction.

The QIM model has also proven its applicability in diverse molecular design tasks by allowing conditional sampling for multi-objective molecular generation and the interpolation of transition pathways of chemical transformations. Unlike the traditional generative paradigm in machine-learned latent spaces, our approach allows to determine and interpret the input conditional distribution since every coordinate

in our model is a QM property with a clear physical meaning. The modified training procedure applied to our model provided an initial assessment for using inverse design to generate de novo molecules with desired QM properties, highlighting the promise of rational exploration of CCS, but also uncovering the relative limitation of generative models in extrapolating to molecules larger than the training samples. Future studies in this direction should investigate the use of both a more refined molecular representation to effectively capture both geometric/electronic features and a dataset spanning a much larger chemical space than the one systematically covered by QM7-X dataset[62]. Our work also suggests that the common latent representation of properties and structures can capture essential aspects of the underlying physics governing structure-property relationships, namely, the existence of transition pathways in chemical space. In this regard, the successful execution of a NEB calculation using the generated transition geometries is a compelling evidence of the meaningful nature of the latent representation as intrinsic coordinates in chemical space. This establishes a connection between our interpolation method and other studies focused on geodesic transition path interpolation[63]. Indeed, we have demonstrated that aiming to parameterize the CCS through the utilization of QM properties can lead to the development of more interpretable models capable of performing a wide range of chemical tasks.

While the QIM model is multitasking and customizable, it still exhibits certain shortcomings in terms of molecular representation and coverage of chemical space that require further investigation to develop a more robust and generalizable model suitable for ML-based screening pipelines within the field of drug discovery (*vide supra*). Once these limitations are relaxed, one could devise the development of an active learning-like method that uses the self-consistency metric based on latent space overlap. This would allow going beyond the chemical and property space spanned by the training set[51]. Another interesting avenue is the development of a QM-informed generative model for molecules with pharmaceutical and biological relevance (e.g., Aquamarine[64], MoleculeNet[65], and solubility[66] datasets) by coupling in-silico (gas-phase and solvated) QM properties with experimental data to exploit unknown property correlations in more realistic scenarios[67]. Hence, we expect this study to serve as a motivation for future research endeavors focused on advancing the field of generative models by leveraging physical and chemical design rules obtained from structure-property/property-property relationships, fostering the development of enhanced models that offer increased accuracy for practical chemical tasks while preserving the high interpretability of the presented approach.

## Methods
### Need for a tailored implementation

To achieve the goals defined in the "Introduction" section, our idea was to look for a generative model that provides a direct differentiable mapping from the space of QM properties to the space of chemical structures. To this end, we first considered the state-of-the-art models in conditional 3D molecular generation that were recently developed such as the equivariant diffusion model (EDM)[43] and the conditional generative neural network (cG-SchNet)[42]. The methodology put forth in EDM, which has been adapted in multiple landmark works[33,44,46,47], is a diffusion-based model that acts on a graph-based representation of molecules. Despite showing good performance in specific applications, the use of a graph-based representation poses a significant limitation for our work. The need to define the number of nodes externally to the model makes any possible property-structure mapping inherently non-differentiable with respect to the number of atoms parameter. On the other hand, cG-SchNet model works in an autoregressive fashion by learning the conditional distribution of the position and type of each atom in a molecule given the position and type of the previously generated ones. The generation process keeps

adding atoms in euclidian 3D space until a stop token is encountered. This method has been successfully used[51] and allows for the generation of molecules with a varying number of atoms. However, the autoregressive nature together with the numerous adjustments in the final cG-SchNet model prevent us from obtaining a one-shot differentiable mapping. Accordingly, we opted for implementing a more flexible and tailored model that combines a VAE architecture to encode the molecular structures (represented as Coulomb matrices) with a property encoder to encode the associated QM properties, see Fig. 1. The QIM model effectively addresses the discussed challenges in terms of molecular size and differentiability. It enables a direct and differentiable parameterization of the chemical compound space (CCS) based on global properties, and it is also highly explainable.

### Variational auto-encoder framework for inverse design

A Variational Auto-Encoder (VAE) is in general comprised of two neural networks called encoder and decoder. The encoder encodes the input in a lower-dimensional latent space representation and the decoder decodes that representation trying to recover the encoded input. The model thus learns a continuous low-dimensional representation capturing the most salient statistical features of the input during the compression process. Whereas, in the decoding process, the decoder learns how to generate samples coherent with the ones in the dataset. Both networks are probabilistic, meaning that they parameterize a probability distribution. This, together with a regularization term forcing the latent distribution to be close to a reference distribution, helps obtain a less sparse internal representation and avoids large portions of the latent space from decoding into invalid outputs. The loss function used to train these models comes from the well-known evidence lower bound (ELBO)[68]. If $\mathbf{x}$ is a sample from a dataset with probability distribution $p(\mathbf{x})$, then the loss reads:

$$\text{loss}_{\text{ELBO}} = D_{\text{KL}}[q_\phi(\mathbf{z}|\mathbf{x})||p(\mathbf{z})] - \mathbb{E}_{q_\phi}[\log(p_\theta(\mathbf{x}|\mathbf{z}))], \quad (1)$$

where $q_\phi(\mathbf{z}|\mathbf{x})$ and $p_\theta(\mathbf{x}|\mathbf{z})$ are the probability distributions parameterized by the encoder and the decoder, respectively. $D_{\text{KL}}$ is the Kullback-Leibler divergence and $p(\mathbf{z})$ is the prior probability distribution of the latent space representation $\mathbf{z}$. Usually, all distributions are chosen to be identically multivariate Gaussian distributed and $p(\mathbf{z}) = \mathcal{N}(\mathbf{0}, \mathbb{I})$. In the training procedure, the encoder learns to encode $x$ in the probability distribution $q_\phi(\mathbf{z}|\mathbf{x})$ from which its latent representation $\mathbf{z}$ is sampled using the well-known reparameterization trick[68]. The divergence term here ensures that the representation stays compact, since the distribution will be as close as possible to the prior distribution $\mathcal{N}(\mathbf{0}, \mathbb{I})$. Then, $\mathbf{z}$ is fed to the decoder which learns to turn this random variable back to $\mathbf{x}$. After training is over, one can sample $\mathbf{z}$ from $\mathcal{N}(\mathbf{0}, \mathbb{I})$, feed it to the decoder, and obtain samples $\mathbf{x}$ with a distribution close to $p(\mathbf{x})$.

In our implementation, we start from a dataset $D = (X, Y)$ where $X$ are the molecular structures and $Y$ are their QM properties. To obtain an inverse mapping $f: Y \to X$, we modified the standard VAE framework by adding a third network parameterizing the probability distribution $p_\psi(\mathbf{z}|\mathbf{y})$. Moreover, we modify the loss function by including a likelihood term (e.g., $-\log(p_\psi(\mathbf{z}|\mathbf{y}))$) in the ELBO loss function, which is now expressed as:

$$\text{loss} = \beta D_{\text{KL}}[q_\phi(\mathbf{z}|\mathbf{x})||p(\mathbf{z})] - \mathbb{E}_{q_\phi}[\log(p_\theta(\mathbf{x}|\mathbf{z}))] - \tau \mathbb{E}_{q_\phi}[\log(p_\psi(\mathbf{z}|\mathbf{y}))], \quad (2)$$

where $\beta$ and $\tau$ are adjustable coefficients introduced as hyperparameters. The training procedure is here similar as described for VAE but, in this case, the variable $\mathbf{z}$ sampled from $q_\phi(\mathbf{z}|\mathbf{x})$ will also maximize the likelihood term $\mathbb{E}_{q_\phi}[\log(p_\psi(\mathbf{z}|\mathbf{y}))]$. This modification ensures the emergence of a common latent representation for both the VAE and the property encoders. After training, one takes samples of $\mathbf{y}$ from the

property space and gets a value for $z$ as the mean of $p_\psi(\mathbf{z}|\mathbf{y})$ that serves as the maximum-likelihood estimator for the latent space representation of the corresponding molecular structure. Subsequently, a molecule $\mathbf{x}$ can be generated, which is expected to possess properties similar to $\mathbf{y}$. A schematic representation of our proposed implementation is shown in Fig. 1.

### Molecular representation and structure retrieval

In recent years, there has been a surge of groundbreaking research in generative modeling with a predominant focus on text-based representations like SMILES[26–40], which encode information on the atom composition and connectivity within a molecule. However, despite its success, this representation limits our ability to study the QM properties of molecular systems. This issue arises because the electronic density, a key quantity in most QM methods, relies on the atom composition and three-dimensional atomic coordinates as input. Accordingly, since we are addressing the inverse mapping between QM properties and 3D structures, we require a different representation that enables us to address both structural and electronic properties. In particular, we need a representation that fulfills the following criteria: firstly, it has to encode the atomic positions and atomic species of a molecule with as many invariances as possible (translational, rotational and permutational). Secondly, it has to show a strong correlation with QM properties. Finally, and most importantly, it has to be invertible, namely that one should be able to retrieve the Cartesian coordinates and atomic species. Neglecting the permutational invariance requirement, we find that a simple yet effective 3D molecular representation is the Coulomb matrix (CM)[69,70]. CM is an elegant and physically-inspired descriptor, which has shown great success in a wide variety of investigations related to molecular property prediction. CM is invariant to translations and rotations, allowing for the retrieval of atomic positions and species of a molecule up to a chirality transformation. This representation is defined as:

$$\mathbf{M}_{ij} = \begin{cases} 0.5 \mathbf{Z}_i^{2.4} & \text{if } i = j. \\ \frac{\mathbf{Z}_i \mathbf{Z}_j}{|\mathbf{r}_i - \mathbf{r}_j|}, & \text{otherwise}. \end{cases} \quad (3)$$

where the indices $i$ and $j$ run over the atoms in the molecule and $\mathbf{Z}_i$ indicates the atomic species. We have chosen to treat the hydrogen (H) atoms implicitly in our work due to several reasons. Primarily, the associated terms in the Coulomb matrix representation pertaining to H atoms are significantly smaller compared to the other atoms in a given system. As a consequence, even small errors in the reconstruction process can lead to disproportionately large changes in the positions of H atoms. By treating the hydrogen atoms implicitly, we can mitigate the impact of these potential errors and minimize the distortion caused by inaccuracies in their positions. This approach allows us to focus our method on accurately representing the molecular scaffold (i.e., positions of heavy atoms). For what concerns the network input standardization, we adapted the matrix to the maximum possible number of atoms per species found in the dataset, padding the rest of the entries with zeros (see Supplementary Note 1 of the Supplementary Information (SI)).

The retrieval of Cartesian coordinates and chemical composition from CM follows two steps. We first obtain the composition from the diagonal elements by applying the inverse function $g = (2(\cdot))^{\frac{1}{2.4}}$ and rounding the outcomes to the closest integer values. Accordingly, we use the set $\{Z\}$ of atomic numbers obtained and get the distances $\mathbf{d}_{ij}$ as $\mathbf{d}_{ij} = \left(\frac{\mathbf{M}_{ij}}{\mathbf{Z}_i \mathbf{Z}_j}\right)^{-1}$. Lastly, we apply classical multidimensional scaling (CMDS) to the resulting Euclidian distance matrix (EDM)[71,72] to get the Cartesian coordinates of atoms up to a chirality transformation (see Supplementary Note 1 of the SI). To reconstruct H atoms, we use OpenBabel

software[73] that restores connectivity and bond order based on interatomic distances. After this, the positions of the H atoms are optimized with third-order self-consistent charge density-functional tight binding (DFTB3)[74,75] that also accounts for many-body dispersion/van der Waals (vdW) interactions *via* the MBD approach[76,77] (DFTB3+MBD, same level of theory used for optimizing the geometries in QM7-X dataset), while freezing the position of the rest of the atoms in the molecule.

As a final remark concerning the choice of representation, a graph-based representation would be beneficial for generative purposes as it treats molecular fragments robustly and avoids the need for truncation. However, in this work, we are aiming at defining a CCS parameterization based on a set of global QM properties of molecules. Consequently, it becomes crucial to account for the potential variation in the number of atoms as we navigate through the property coordinates. Given the challenges associated with graph-based approaches due to the requirement of specifying the number of nodes in advance, the utilization of (appropriately padded) CM proves to be an efficient and scalable representation for inverse design purposes at this stage of model development.

## Reference chemical space
Out of the ~ 4.2M molecules in the QM7-X dataset[54], we have selected a subset of 40, 988 equilibrium conformations for molecules containing up to seven heavy atoms, including C, N, and O. For training our model, we used the following data splitting: 28k and 2k molecules for the training and validation sets, respectively, while the remaining molecules were used for testing the model. Moreover, since the QM7-X dataset contains a plethora of physicochemical properties, computed by means of non-empirical hybrid density-functional theory (DFT) and a many-body treatment of vdW/dispersion interactions (i.e., PBE0+MBD)[76,78–80] in conjunction with the tightly converged numeric atom-centered basis sets[81], we have opted to consider 17 global (extensive and intensive) properties during the training process.

## Gradient attribution map
Similar to the standard approach employed in machine learning applications that involve image inputs[82], we compute the gradient of the Coulomb matrix components $\mathbf{CM}_i^k$ of a given molecule $k$ with respect to a property $\mathbf{p}_j$, yielding the Jacobian matrix $\mathbf{J}_{ij}^k = \frac{\partial \mathbf{CM}_i^k}{\partial \mathbf{p}_j}$. By taking the norm of $\mathbf{J}_{ij}^k$ over the output dimension of $\mathbf{CM}$ and, then, averaging over the subset $\mathbb{B}$ of best reconstructed molecules (150 molecules with RMSD ≤0.2 Å), we obtain an attribution map for each property expressed as:

$$\mathbf{A}_j = \frac{1}{N}\sum_{k\in\mathbb{B}}\left\|\frac{\partial\mathbf{CM}_i^k}{\partial\mathbf{p}_j}\right\|, \qquad (4)$$

where $N$ is the total number of molecules included in $\mathbb{B}$. The idea behind this calculation is to understand which changes in the input produce the largest changes in the output.

## Multi-Gaussian fitting for conditional molecular design
Here, we have used a multi-Gaussian fitting procedure to show the capability of the QIM model to target predefined pairs of QM properties, denoted as {$\mathbf{m}$}. In doing so, we first acknowledge that our model has a fixed-dimensional input space for properties, ensuring the attainment of the random sampling associated with typical generative models. The conditional sampling is then achieved by sampling the non-targeted properties, denoted as {$\mathbf{n}$}, while conditioning on {$\mathbf{m}$} values. For this purpose, we need a modeled version of the distribution of the property space spanned by QM7-X that facilitates the determination of the posterior distribution of {$\mathbf{n}$}, given fixed {$\mathbf{m}$} values. Indeed, we use a multivariate Gaussian regression approach to model this property space. Specifically, we constructed a model with 91 multivariate Gaussian

distributions $\{\mathcal{N}(\boldsymbol{\mu}_k,\boldsymbol{\Sigma}_k)\}_{k\in\{1[...]91\}}$ with $\boldsymbol{\mu}_k$ and $\boldsymbol{\Sigma}_k$ as the mean value and the covariance matrix of the Gaussian $k$. This choice was made following a Bayesian information criterion (see Supplementary Note 4 of the SI).

With fixed target values, denoted as {$\bar{\mathbf{m}}$}, the Gaussian $\bar{k}$ from which it is more likely to sample the targeted properties is then selected following the maximum likelihood criterion $\bar{k} = \text{argmax}k(\mathcal{N}_{\mathbf{m}}(\mathbf{m} = \bar{\mathbf{m}}|\boldsymbol{\mu}_k,\boldsymbol{\Sigma}_k))$, where $\mathcal{N}_{\mathbf{m}}$ is the marginal distribution relative to the chosen set of properties {$\mathbf{m}$}. Next, the conditional probability formula for multivariate Gaussian distributions is applied, obtaining the distribution of the non-targeted properties {$\mathbf{n}$} for $\mathbf{m} = \bar{\mathbf{m}}$. This can be written as,

$$p(\mathbf{n}|\mathbf{m} = \bar{\mathbf{m}}) = \mathcal{N}(\mathbf{n}|\tilde{\boldsymbol{\mu}},\tilde{\boldsymbol{\Sigma}}), \qquad (5)$$

with $\tilde{\boldsymbol{\mu}}$ and $\tilde{\boldsymbol{\Sigma}}$ defined as,

$$\tilde{\boldsymbol{\mu}} = \boldsymbol{\mu}_n + \boldsymbol{\Sigma}_{nm}\boldsymbol{\Sigma}_{mm}^{-1}(\bar{\mathbf{m}} - \boldsymbol{\mu}_m) \qquad (6)$$

$$\tilde{\boldsymbol{\Sigma}} = \boldsymbol{\Sigma}_{nn} - \boldsymbol{\Sigma}_{nm}\boldsymbol{\Sigma}_{mm}^{-1}\boldsymbol{\Sigma}_{mn}. \qquad (7)$$

To proceed with the generation of molecular structures, multiple values for {$\mathbf{n}$} are sampled from $\mathcal{N}(\mathbf{n}|\tilde{\boldsymbol{\mu}}(\mathbf{m}),\tilde{\boldsymbol{\Sigma}}(\mathbf{m}))$ for a fixed values of the targeted properties $\mathbf{m} = \bar{\mathbf{m}}$ (the variables with "$\bar{\phantom{m}}$" on top correspond to the target values throughout the text). Each sample $s$ will correspond to the total set of (targeted and non-targeted) properties $\bar{\mathbf{m}} \oplus \mathbf{n}_s$ which is then fed to the model that will generate a Coulomb matrix sample $\mathbf{CM}_s$. At this stage, the VAE encoder is used to encode the samples and, consequently, to obtain the latent representation encoding $\mathbf{z}_s$. This is used to filter the samples based on the self-consistency criterium from defined when analyzing the model's performance. Namely we consider $\Delta\mathbf{z} = ||\mathbf{z} - \mathbf{z}_s|| \in [0,0.4]$, where $\mathbf{z}$ is the encoding from the property encoder. Note that this specific interval is adjusted for each target, taking into consideration the possibility of it being either too loose or excessively stringent. Cartesian coordinates, atomic species of molecules and H atoms are posteriorly retrieved from the generated Coulomb matrices as described above. After checking for broken molecules, we optimize the position of H atoms and then screen for overlapping atoms by putting a threshold of 10 a.u. to the maximum force component at with DFTB3+MBD level of theory. As a final step, the generated structures undergo geometry optimizations using DFTB3+MBD and, subsequently, their QM properties are calculated at the PBE0+MBD level of theory to enable a comparison with the target values {$\bar{\mathbf{m}}$} in the QM7-X dataset.

## Masking method for representation
Stemming from the consideration that the padded Coulomb matrix representation presents issues in treating molecular fragments independently, we decided to address the problem during the training procedure with the following modifications:

- Randomly masking atoms: This is done by taking a random binary vector of length $l$, where the padded Coulomb Matrix is an $l \times l$ matrix, and by taking the outer product of the said vector. The distribution of zeros and ones is a Bernoulli distribution with $p = 0.5$. To maintain the same expected value for the reconstruction loss, we also multiply the mask by a factor 2. The resulting mask $\mathcal{M}$ is then applied to the reconstruction loss function before taking the sum over the Coulomb matrix, namely:

$$loss_{\mathcal{M}} = \Sigma_{i,j}\mathcal{M}_{ij}(-\log(p_\theta(\mathbf{x}_{ij}|\mathbf{z}))), \qquad (8)$$

where $loss_{\mathcal{M}}$ is the new reconstruction loss.

- Leaky masking of padding: this is done in a similar way by setting $\mathcal{M}_{ij}$ to 0.1 if $\mathbf{x}_{ij} = 0$.

The application of this masking significantly enhances the capability of QIM model to treat fragments independently. Additionally, due to the leaky masking of the padding, the model can consider molecules with more heavy atoms than the largest molecule in the dataset. To generate the unseen compounds, we have repeated the initial procedure without using the self-consistency filter and generating samples until a new heavy atom composition is found.

## Geodesic interpolation algorithm

While it would be relatively straightforward to perform a linear interpolation in the latent space and generate associated geometries using the learned CCS parameterization, we are also interested in exploring the feasibility of obtaining an energy profile estimation along with this task. To do so, we use the geodesic interpolation algorithm[58] for VAEs but, in this instance, the objective is to find curves in the property space that are geodesics with respect to the metric induced by the latent space encoding. This enables us to see how a close-to-linear interpolation in the latent space is reflected in both the property input space and the structure output space. The procedure to get the transition geometries consists in optimizing with a gradient descent algorithm an initial linear interpolation in property space between two structures, by minimizing the distance of the path in latent space.

The geodesic interpolation procedure starts with considering the initial and final conformational isomers, namely $\mathbf{x}_0$ and $\mathbf{x}_N$. We then linearly interpolate $N-1$ points between the corresponding coordinates in property space defined by the seventeen properties considered during the training procedure. Applying the property-to-structure relationship defined by our model, we obtain an initial configuration given by the set $\{\mathbf{y}_0, \mathbf{y}_1, \mathbf{y}_2, ..., \mathbf{y}_N\}$ of property coordinates, the set $\{\mathbf{z}_0, \mathbf{z}_1, \mathbf{z}_2, ..., \mathbf{z}_N\}$ of latent space coordinates and the set $\{\mathbf{x}_0, \mathbf{x}_1, \mathbf{x}_2, ..., \mathbf{x}_N\}$ of molecular structures. After this, we use a gradient descent algorithm to minimize the loss function $L = \Sigma_{i=1}^{N}||\mathbf{z}_{i+1} - \mathbf{z}_i||^2 + \epsilon\Sigma_{i=1}^{N}||\mathbf{y}_{i+1} - \mathbf{y}_i||^2$, where the first term enforces the minimum distance in the latent space, while the second term is a regularization term with $\epsilon << 1$ to enforce continuity in the property space. To find a stationary path, the optimization runs until a convergence criterion on the norm of the gradient is met (i.e., $< 10^{-4}$). Concerning the choice of molecules for this evaluation test, we also took into account that chiral molecules are indistinguishable when using the CM representation. We hence select the initial and final states ensuring that these are not mirror images of one another, as the model would interpret them as identical molecules.

## Reporting summary

Further information on research design is available in the Nature Portfolio Reporting Summary linked to this article.

## Data availability

The QM7-X data[54] that support the findings of this study are available in ZENODO with the identifier https://doi.org/10.5281/zenodo.4288677. The preprocessed data to make our implementation work are provided in the Github repository Repo-AleFalla[83]. Source data are provided with this paper. The authors declare that all data supporting the findings of this study are available within the paper and its supplementary information files. Source data are provided with this paper.

## Code availability

The code corresponding to our implementation is available in the Github repository Repo-AleFalla[83]. Moreover, we provide a Jupyter notebook to reproduce the main results.

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

## Acknowledgements

We thank Dr. Szabolcs Goger and Mr. Kyunghoon Han for the helpful discussions and comments to improve the manuscript. This research was financially supported by the European Union's Horizon 2020 research and innovation program under the Marie Skłodowska-Curie Innovative Training Network - European Industrial Doctorate grant agreement No 956832, "Advanced Machine learning for Innovative Drug Discovery" (AIDD). The results discussed in this work were obtained using the computational resources provided by the High Performance Computing (HPC) at the University of Luxembourg.

## Author contributions

The work was initially conceived by A.F. and L.M.S., and designed with contributions from A.T. A.F. developed the machine learning code, performed the model training, and analyzed the model performance in diverse applications together with L.M.S. A.T. supervised and revised all stages of the work. All authors discussed the results and contributed to the final manuscript.

## Competing interests

The authors declare no competing interests.
