## [Peer Review File · Nature Communications]

Inverse Mapping of Quantum Properties to Structures for Chemical Space of Small Organic MoleculesREVIEWER COMMENTS

Reviewer #1 (Remarks to the Author):

The authors of this paper have predicted and designed organic materials with desired properties using a modified Variational Auto-Encoder (VAE). Traditional VAEs design molecules by using an encoder to predict properties from the molecular structures of organic materials and a decoder to predict structures from these properties. However, in this study, by modifying the widely-used ELBO loss function, the authors introduced an additional property encoder that compresses 17 different QM properties into a latent representation space. This allows more efficient multi-objective design of molecular structures. Especially, the NEB/transition geometry result was really interesting. While the research introduces a compelling approach to material discovery, there are potential areas of enhancement, as discussed in the subsequent comments.

(1) While I acknowledge that the concept introduced in this study surpasses the existing approaches, from a mathematical perspective, the method presented seems limited to interpolation tasks. It may not be suitable for identifying new chemical compounds whose QM properties exceed the range observed in the training set's molecules. A discussion on enabling extrapolation tasks would be an intriguing addition.

(2) Upon reviewing Figure 3 and Figure 6, as well as the conclusion, it seems that extensive properties play a significant role in determining the latent representation and molecular design. In contrast, the influence of intensive properties appears to be minimal, making their accurate prediction or design challenging. I'm curious if, in cases where intensive properties are crucial, predicting and designing the latent representation using only intensive properties might enhance performance.

(3) In Figure 2 (c), it appears that there is a disconnected linkage within the atoms of the reconstructed molecule in the third box. I'm curious about how you handled atomic connections when obtaining coordinates by inverting the Coulomb matrix, and whether this disconnection is an oversight or an error. Additionally, it would be helpful if you could annotate which atom each color represents in this figure and subsequent ones.

(4) For practical real-world applications of this method, it's essential to account for the systematic discrepancies between molecular properties derived from computational simulations and those obtained experimentally. Given that the VAE model is currently trained using a computational dataset, the molecular design hinges on properties calculated computationally. How might one approach designing molecules that align with the desired experimental or 'real' properties?

Reviewer #2 (Remarks to the Author):

The presented work introduces a pioneering machine-learning approach to the challenging task of inverse property-to-structure design for small organic molecules. By learning a parameterization of the chemical compound space (CCS) using Quantum Mechanics (QM) properties as intrinsic coordinates, the authors have developed a proof-of-concept implementation that combines a variational autoencoder and a property encoder. I would like to commend the authors for their intriguing work. However, I would appreciate additional clarification on certain aspects of the presented content before publishing it in Nature Communications. Here are my queries and suggestions:

1. Concerning the modified loss function in the section Variational Auto-Encoder framework for inverse design, the authors introduce adjustable coefficients β and τ . Could there be a detailed explanation of how these coefficients were chosen and their specific roles in the model training process?

2. The use of the Coulomb matrix (CM) is justified based on its elegance and success in molecular property prediction. Could the authors provide more insights into why CM is considered physically inspired and how it achieves invariance to translations and rotations?
3. Treating hydrogen (H) atoms implicitly is a notable choice. Could the authors elaborate further on why this approach is chosen and how it impacts the accuracy of the representation?
4. The use of OpenBabel software for reconstructing H atoms is mentioned. Could the authors provide more details on how this software restores connectivity and bond order based on interatomic distances?
5. The authors mention that a graph-based representation would be beneficial for generative purposes but chose CM for this work. Could the authors discuss the trade-offs and considerations that led to this choice, especially in the context of defining a CCS parameterization based on global QM properties?
6. The empirical threshold of $\text{RMSD} = 0.7 \text{ \AA}$ is mentioned as adequate for separating molecules with acceptable reconstruction. Could the authors discuss how this threshold was determined and its significance in the context of heavy atom structure reconstruction?
7. The concept of the gradient attribution map is intriguing. Could the authors provide a brief explanation or context on how the gradient attribution map aids in evaluating the contribution of each property to the output structures?
8. The correlation of A values with two-dimensional PCA of the latent space is discussed. Could the authors provide additional insights into why certain QM properties exhibit higher correlation with PCA, and how this knowledge contributes to the interpolation of transition structures?
9. The distribution of molecular structures based on different pairs of QM properties with high and low A values is discussed. Could the authors provide more insights into the practical implications of these distributions and how they contribute to identifying molecular isomer subspaces?
10. It's noted that the modification slightly reduces the model performance in molecular reconstruction. Could the authors elaborate on the trade-offs made by reducing bias and how it affects the model's overall performance? Are there specific aspects of molecular reconstruction that are more affected than others?
11. The EMBD and α values for the generated molecules are discussed in the context of the expected range for molecules with eight heavy atoms. Could the authors provide additional insights into how well the generated molecules align with the expected values, and whether any outliers or anomalies were observed?
12. The manuscript concludes by acknowledging the relative limitation of generative models in extrapolating to molecules larger than the dataset. Could the authors elaborate on potential strategies or directions for future studies to overcome this limitation, especially regarding the use of advanced representations and more extensive datasets?

Reviewer 1

The authors of this paper have predicted and designed organic materials with desired properties using a modified Variational Auto-Encoder (VAE). Traditional VAEs design molecules by using an encoder to predict properties from the molecular structures of organic materials and a decoder to predict structures from these properties. However, in this study, by modifying the widely-used ELBO loss function, the authors introduced an additional property encoder that compresses 17 different QM properties into a latent representation space. This allows more efficient multi-objective design of molecular structures. Especially, the NEB/transition geometry result was really interesting. While the research introduces a compelling approach to material discovery, there are potential areas of enhancement, as discussed in the subsequent comments.

Our reply: We thank the reviewer for the favorable assessment of our work. Detailed point-by-point responses to each of the reviewer's comments can be found below.

(1) While I acknowledge that the concept introduced in this study surpasses the existing approaches, from a mathematical perspective, the method presented seems limited to interpolation tasks. It may not be suitable for identifying new chemical compounds whose QM properties exceed the range observed in the training set's molecules. A discussion on enabling extrapolation tasks would be an intriguing addition.

Our reply: We thank the reviewer for his/her comment. We agree with the reviewer that our current implementation struggles with extrapolation outside the property range of the training set's molecules. However, there seem to be ways to mitigate this such as the one we implemented in our work, see Sec. 4 of the Supplementary Information (SI). We modified the conclusions, now named "Discussion", to include in more detail the considerations about representation and coverage of the chemical space, but more importantly, potential research directions that might help move away from the initial property distribution considered. See text highlighted in blue on "Discussion" section in the main text.

(2) Upon reviewing Figure 3 and Figure 6, as well as the conclusion, it seems that extensive properties play a significant role in determining the latent representation and molecular design. In contrast, the influence of intensive properties appears to be minimal, making their accurate prediction or design challenging. I'm curious if, in cases where intensive properties are crucial, predicting and designing the latent representation using only intensive properties might enhance performance.

Our reply: We thank the reviewer for his/her suggestion. While the reviewer is correct in stating that intensive properties may not play as significant a role as extensive ones in structure reconstruction, we have found that including intensive properties can further improve the performance of our model (now named as QIM model, which stands for "Quantum Inverse Mapping"). In Fig. R1 (also Fig. S2 of the SI), one can see that there still is an increment in the percentage of well-reconstructed molecules below the RMSD threshold of 0.7 Å when going from 8 to 17 properties considered in the training procedure. Following the suggestion from the reviewer, we have also trained two additional models to analyze the influence of each property type, *i.e.*, we now have a model trained only on extensive properties and another one trained only on intensive properties. In Fig. R2 (also Fig. S4 of the SI), we show the RMSD histogram and its corresponding cumulative function for the models trained using the set of 17 QM properties (blue), only extensive properties (green), and only intensive ones (orange). These results verify the need to use both types of properties in the training of our final model to have a better reconstruction of molecular structures. The models trained on separated properties present a higher number of structures with large RMSD compared to the final model, indicating a lower performance in structure reconstruction. Moreover, the model trained only on intensive properties was able to reconstruct with the right chemical composition approximately 8,000 from 10,000 molecules of the test set, see Fig. R2(b). In brief, this analysis has confirmed that considering the entire subset of 17 quantum-mechanical properties yields

Figure R 1 Cumulative distribution function for RMSD computed using molecules from the test and different numbers of QM properties in the reconstruction procedure.

Figure R 2 a) Histograms and b) cumulative distribution function of the RMSD for molecules from the test set in molecular reconstruction evaluation when considering all properties, only intensive properties, and only extensive properties in the training procedure. In either case, the counts are scaled to show the difference in performance for composition reconstruction.

the best performance in the present work. This discussion has been added to the main text in the “Results” and “Discussion” sections as well as to Sec. 1.2 of the SI.

(3) In Figure 2(c), it appears that there is a disconnected linkage within the atoms of the reconstructed molecule in the third box. I’m curious about how you handled atomic connections when obtaining coordinates by inverting the Coulomb matrix, and whether this disconnection is an oversight or an error. Additionally, it would be helpful if you could annotate which atom each color represents in this figure and subsequent ones.

Our reply: We thank the reviewer for his/her observation. In this work, the 3D molecular structures are reconstructed from the predicted values of the Coulomb Matrix representation using the method explained in Sec. 1 of the SI. This method only provides atomic positions and chemical composition of molecules rather than

connectivity. The observed disconnected linkage between the atoms in Figure 2(c) results from the counting of valence electrons and longer interatomic distances compared to the predefined set of equilibrium bond lengths used in the visualizer for plotting the molecules. We have added the information about the color code of atoms in the caption of Figure 2 of the main text.

(4) For practical real-world applications of this method, it's essential to account for the systematic discrepancies between molecular properties derived from computational simulations and those obtained experimentally. Given that the VAE model is currently trained using a computational dataset, the molecular design hinges on properties calculated computationally. How might one approach designing molecules that align with the desired experimental or 'real' properties?

Our reply: We thank the reviewer for his/her comment. Over the last few years, there has been an increase in investigations attempting to relate quantum-mechanical (QM) data to the chemical and biological properties of organic systems of pharmaceutically relevant size and composition. In this regard, QM datasets of gas-phase and solvated large drug-like molecules have been generated and are waiting to be used as a benchmark in state-of-the-art generative models. Building on this idea, we have included a statement in the "Discussion" section regarding a potential research direction that aims to exploit the unknown correlations between QM properties and experimental data for molecular generation: *"Another interesting avenue is the development of QM-informed generative model of molecules with pharmaceutical and biological relevance (e.g., Aquamarine, MoleculeNet, and solubility datasets) by coupling in-silico (gas-phase and solvated) QM properties with experimental data to exploit unknown property correlations in more realistic scenarios."*

Reviewer 2

The presented work introduces a pioneering machine-learning approach to the challenging task of inverse property-to-structure design for small organic molecules. By learning a parameterization of the chemical compound space (CCS) using Quantum Mechanics (QM) properties as intrinsic coordinates, the authors have developed a proof-of-concept implementation that combines a variational autoencoder and a property encoder. I would like to commend the authors for their intriguing work. However, I would appreciate additional clarification on certain aspects of the presented content before publishing it in Nature Communications. Here are my queries and suggestions:

Our reply: We thank the reviewer for the favorable assessment of our work. Detailed point-by-point responses to each of the reviewer’s comments can be found below.

1. Concerning the modified loss function in the section Variational Auto-Encoder framework for inverse design, the authors introduce adjustable coefficients β and τ . Could there be a detailed explanation of how these coefficients were chosen and their specific roles in the model training process?

Our reply: The role of the parameters β and τ is to tune and regularize the feature extraction process and the latent space organization, giving initially more weight to latent space compression and later to prediction from properties. The idea behind this scheduling is that in the early stages of training, we want to obtain an initial compact representation, which makes a good starting point for the training of the VAE component of the implementation, but also makes it easy for the latent space predictor to initially have a low loss. After this initial phase, we want the latent space to start being more expressive. This should happen slowly enough so that the latent space predictor can keep up with the changes in the internal representation of the VAE. Meanwhile, for the VAE decoder, it will be increasingly easier to reconstruct the original molecule.

Fig. R3 (also Fig. S5 of the Supplementary Information (SI)) shows the influence of the selection of the β parameter on the learning process. Starting from $\beta = 3$, we can observe that the KLD component of the loss is minimized, implying a more compact latent space, while the reconstruction loss of the latent representation is low (as expected from the first earlier considered effect). This finding, along with a higher value of the molecule reconstruction loss, indicates a less expressive latent representation. For $\beta = 1$, it can be observed from the KLD that the latent space never compactifies as much as in the previous case, leading to a worse performance in the latent space predictor. The scheduling with decaying β provides instead, as we wanted, an initial phase of latent space compression, where latent reconstruction loss is still as low as in the $\beta = 3$ case. Then, we see an increase of the KLD which, from being lower than the plateau from the $\beta = 1$ case in the first phase, goes back up and plateaus close to the value for $\beta = 1$. For the molecule reconstruction loss, the use of this scheduling results in a more expressive latent representation that has learned to capture relevant features during the compression phase. Moreover, the higher value of KLD is a confirmation that the low loss in latent space reconstruction does not come from a collapse of the internal representation towards the uninformative prior $\mathcal{N}(0, \mathbb{I})$. Fig. R3(e) shows a summary of these results by considering only the reconstruction components of the loss for all the considered schedules. Finally, regarding the choice of the τ parameter, we experimented with various values within the range of (1,3). This task led us to the determination that $\tau = 2$ was the optimal choice, despite observing the minimal impact on the final performances in terms of $\langle RMSD \rangle$, see Fig. R4 (also Fig. S6 of the SI). The discussion of the meaning and selection of both parameters has been added to Sec. 2 of the SI.

2. The use of the Coulomb matrix (CM) is justified based on its elegance and success in molecular property prediction. Could the authors provide more insights into why CM is considered physically inspired and how it achieves invariance to translations and rotations?

Our reply: The Coulomb matrix (CM) is a physically inspired representation with origins in Coulomb’s law, which describes the pairwise electrostatic interaction between electrically charged particles. However, to construct the CM representation, the atomic numbers Z are considered instead of atomic charges, see Eq. (3)

Figure R 3 Variation of loss components during the training process considering different schedulings of the β parameter. a) Reconstruction loss latent, b) reconstruction loss molecule, c) KLD component, d) β parameter, and e) total reconstruction loss.

of the main text. A self-interaction term, which only depends on the atomic numbers, was also considered in its construction. As the CM representation is based on pairwise distances, the resulting matrix is inherently rotation and translation invariant. We are referring the reader to the references where the CM representation was proposed.

3. Treating hydrogen (H) atoms implicitly is a notable choice. Could the authors elaborate further on why this approach is chosen and how it impacts the accuracy of the representation?

Our reply: The choice of treating hydrogen atoms implicitly is mainly because the magnitude of Coulomb Matrix (CM) elements associated with hydrogens is much smaller than any other entry. Most of the time, it is smaller than the reconstruction error, making this extra information mere noise that may deteriorate the performance of the final model.

4. The use of OpenBabel software for reconstructing H atoms is mentioned. Could the authors

Figure R 4 \langle RMSD \rangle obtained by the trained model with different τ values. We have used a fixed number of 1000 epochs to train these models.

provide more details on how this software restores connectivity and bond order based on interatomic distances?

Our reply: The restoration of bond order in OpenBabel is based on interatomic distance and atom type. This information is utilized by the software, which adds hydrogen atoms to satisfy the valency of each atom. For more information about this procedure, we are referring the reader to the software documentation and the corresponding publication.

5. The authors mention that a graph-based representation would be beneficial for generative purposes but chose CM for this work. Could the authors discuss the trade-offs and considerations that led to this choice, especially in the context of defining a CCS parameterization based on global QM properties?

Our reply: To clarify and give more details about the choice of our proof-of-concept implementation, we have added the subsection ‘‘Need for a tailored implementation’’ in the ‘‘Methods’’ section. In brief, our idea was to look for a generative model that provides a direct differentiable mapping from the space of QM properties to the space of chemical structures. While state-of-the-art graph-based generative models show good performance in targeting design, their construction makes any possible property-to-structure mapping inherently non-differentiable with respect to the number of atoms parameter. Accordingly, we opted for implementing a more flexible and tailored model that combines a VAE architecture to encode the molecular structures (represented as Coulomb matrices) with a property encoder to encode the associated QM properties.

6. The empirical threshold of $\text{RMSD} = 0.7 \text{ \AA}$ is mentioned as adequate for separating molecules with acceptable reconstruction. Could the authors discuss how this threshold was determined and its significance in the context of heavy atom structure reconstruction?

Our reply: We thank the reviewer for his/her suggestion. As discussed in the main text, the threshold of 0.7 \AA is an empirical value based on how well the model reconstructs molecular topology. We found that there is a general deterioration in the reconstruction of the topology of molecular structures as well as broken molecules after this threshold. To quantify this effect, we now plot the percentage of molecules with correctly

Figure R 5 a) Percentage of reconstructed molecules with correct topology up to a given RMSD value, e.g., when considering molecules with $\text{RMSD} < 0.4 \text{ \AA}$ then 80% of those are reconstructed with correct topology. Distributions of the topological errors fraction (correct bonds over the total number of bonds in the molecule) for RMSD = a) 0.6 \AA , b) 0.7 \AA , c) 0.8 \AA .

reconstructed topology with RMSD values below a given threshold, see Fig. R5 (also Fig. S3 of the SI). We have added a detailed explanation of these results in Sec. 1.2 of the SI, and this section has been referred in the main text.

7. *The concept of the gradient attribution map is intriguing. Could the authors provide a brief explanation or context on how the gradient attribution map aids in evaluating the contribution of each property to the output structures?*

Our reply: The gradient attribution map is a concept inspired by a method used in machine learning for images called *Saliency map*. The idea is to understand which changes in the input produce the largest changes in the output. This is achieved by using the gradient norm as a means to estimate the magnitude of the inverse mapping slope for each specific property. We have added the reference to the original paper where this concept was first introduced and slightly modified the description of the gradient attribution map calculation in the “Methods” section.

8. *The correlation of A values with two-dimensional PCA of the latent space is discussed. Could the authors provide additional insights into why certain QM properties exhibit higher correlation with PCA, and how this knowledge contributes to the interpolation of transition structures?*

Our reply: The reasons for the high correlation of certain properties with PCA are the same as those for finding extensive end energy-related properties to be more relevant than others. In particular, as we mentioned when discussing the CM representation, the extensive character of properties is largely connected to the number of atoms. Changes in that variable translate to having more or fewer non-zero rows and columns in the CM representation. Moreover, the representation of molecules with interatomic distances will increase the

dependency on energetic properties, since they also depend on chemical bonds. The connection between the high correlation of certain properties with PCA and the chemical tasks of interpolating transition states is now better explained at the beginning of the “Interpolating transition geometries between isomers” subsection in the main text. It reads as: *“Indeed, it was demonstrated that the latent space of the QIM model exhibits a discernible structure characterized by energetic properties with high A values when undergoing a linear transformation such as PCA, see Fig. 3. This particular behavior assures that a linear interpolation in latent space will generate structures for which the energetic properties smoothly change—a consequence of the fact that linear transformations do not affect convexity properties up to a sign.”*

9. *The distribution of molecular structures based on different pairs of QM properties with high and low A values is discussed. Could the authors provide more insights into the practical implications of these distributions and how they contribute to identifying molecular isomer subspaces?*

Our reply: In this work, we have found that extensive properties present higher A values compared to intensive ones, *i.e.*, they are more relevant for molecular reconstruction. This is a result of the purely geometric descriptor used in the training procedure of our model, as discussed throughout the manuscript. Accordingly, extensive properties serve as efficient local coordinates for identifying isomer subspaces, since they strongly depend on the total number of atoms and the heavy atom composition—parameters necessary for characterizing isomers. On the contrary, intensive properties have a strong dependence on the atom connectivity and, hence, these properties can largely differ between isomers with different bond arrangements, see Fig. 4(d) of the main text. These insights could be utilized to conditionally generate molecules belonging to specific isomer subspaces within property spaces defined by more pertinent quantum-mechanical properties such as the ones investigated in Fig. 6 of the main text. We have added this statement to the “Discussion” section in the main text.

10. *It’s noted that the modification slightly reduces the model performance in molecular reconstruction. Could the authors elaborate on the trade-offs made by reducing bias and how it affects the model’s overall performance? Are there specific aspects of molecular reconstruction that are more affected than others?*

Our reply: We thank the reviewer for his/her observation. The differences between model performance before and after applying the new masking method are not strong. We obtained that the average performance in molecular reconstruction measured by $\langle \text{RMSD} \rangle$ is only reduced by 0.05 Å. As a result, there is a shift of molecules in the RMSD distribution towards worse reconstruction. This effect happens because the masking method described in Sec. 4 of the SI gives a low weight to padding values which in turn diminishes the model’s ability to discern which entries of the generated samples should be kept to zero. We have added this information in the “Results” section.

11. *The EMBD and α values for the generated molecules are discussed in the context of the expected range for molecules with eight heavy atoms. Could the authors provide additional insights into how well the generated molecules align with the expected values, and whether any outliers or anomalies were observed?*

Our reply: As demonstrated in the revised manuscript, we successfully generated molecules larger than those present in the training set (*i.e.*, molecules with up to seven heavy atoms) with desired QM properties by applying a new masking method during the training procedure, see Sec. 4 of the SI. For this application test, we restricted our molecular generation pipeline to output only molecules with more than seven heavy atoms. Despite the slight increase in the error for property prediction, all generated molecules for both targeted pairs of QM properties were well-aligned with chemical and physical intuition, *i.e.*, neither outliers nor anomalies were observed. Indeed, when comparing Fig. 6(a) and Fig. 7 in the main text, one can see that the molecule generated in the target T_2 is larger and more compact than molecules in target A. Consequently, this molecule will present a higher α and a larger $|E_{MBD}|$, as shown in Fig. 7. Likewise, molecules generated in target T_1 exhibit a larger $|E_{MBD}|$ than those in target B due to the more complex heavy atom composition and compact structure. We have added this information in the “Results” section.

12. *The manuscript concludes by acknowledging the relative limitation of generative models in extrapolating to molecules larger than the dataset. Could the authors elaborate on potential strategies or directions for future studies to overcome this limitation, especially regarding the use of advanced representations and more extensive datasets?*

Our reply: We have added different statements related to these topics in the “Discussion” section, see text highlighted in blue.

REVIEWER COMMENTS

Reviewer #1 (Remarks to the Author):

The authors have adequately addressed the concerns raised by the reviewer, and I support the publication of the revised manuscript.

Reviewer #2 (Remarks to the Author):

This paper highlights the challenge of the inverse mapping in chemical space despite direct mappings from molecular structures to properties using quantum-mechanical methods and machine learning. The authors propose parametrizing chemical space with a finite set of quantum mechanical properties, introducing the QIM model for property-to-structure mapping. Validated on small drug-like molecules, the approach demonstrates explainability and the generation of novel molecular structures with targeted properties. Overall, this work offers a promising proof-of-concept for inverse property-to-structure design across diverse chemical spaces. I have some minor suggestions as discussed below:

1. The authors use the Coulomb matrix (CM) as the material representation, considering the potential variation in the number of atoms and favoring it over graph-based representations. However, besides graph-based approaches, string-based representations like SMILES are also prevalent in materials representation and inverse problems, particularly with the rise of generative large language models. Therefore, expanding the discussion on material representation choice on page 13, lines 377-383, to include such alternatives could enrich the analysis.
2. In Fig.3-7, the author extensively utilizes principal component analysis (PCA), showcasing commendable interpretability and insights into the structure-property-relationship and variance distribution. However, in certain analysis scenarios unrelated to the material properties, such as evaluating the quality of generative models in Fig.5, there might be a greater interest in the local structure and clustering similarity of the data, alternative clustering methods like t-SNE may be more appropriate.

Reviewer #2 (Remarks on code availability):

The authors provide detailed installation instructions and code explanations, and the model is reproducible.

Reviewer 1

The authors have adequately addressed the concerns raised by the reviewer, and I support the publication of the revised manuscript.

Our reply: We thank the reviewer for his/her positive evaluation of the revised version of our manuscript and for recommending publication in *Nature Communications*.

Reviewer 2

This paper highlights the challenge of the inverse mapping in chemical space despite direct mappings from molecular structures to properties using quantum-mechanical methods and machine learning. The authors propose parametrizing chemical space with a finite set of quantum mechanical properties, introducing the QIM model for property-to-structure mapping. Validated on small drug-like molecules, the approach demonstrates explainability and the generation of novel molecular structures with targeted properties. Overall, this work offers a promising proof-of-concept for inverse property-to-structure design across diverse chemical spaces. I have some minor suggestions as discussed below:

Our reply: We thank the reviewer for the favorable assessment of our work. Detailed point-by-point responses to each of the reviewer's comments can be found below.

1. The authors use the Coulomb matrix (CM) as the material representation, considering the potential variation in the number of atoms and favoring it over graph-based representations. However, besides graph-based approaches, string-based representations like SMILES are also prevalent in materials representation and inverse problems, particularly with the rise of generative large language models. Therefore, expanding the discussion on material representation choice on page 13, lines 377-383, to include such alternatives could enrich the analysis.

Our reply: Following the suggestion made by the reviewer, we have expanded the discussion on representation choice. In the Methods section, at the beginning of **Molecular representation and structure retrieval**, you can now find the text: "*In recent years, there has been a surge of groundbreaking research in generative modeling with a predominant focus on text-based representations like SMILES, which encode information on the atom composition and connectivity within a molecule. However, despite its success, this representation limits our ability to study the QM properties of molecular systems. This issue arises because the electronic density, a key quantity in most QM methods, relies on the atom composition and three-dimensional atomic coordinates as input. Accordingly, since we are addressing the inverse mapping between QM properties and 3D structures, we require a different representation that enables us to address both structural and electronic properties.*". The text has been highlighted in red.

2. In Fig.3-7, the author extensively utilizes principal component analysis (PCA), showcasing commendable interpretability and insights into the structure-property-relationship and variance distribution. However, in certain analysis scenarios unrelated to the material properties, such as evaluating the quality of generative models in Fig.5, there might be a greater interest in the local structure and clustering similarity of the data, alternative clustering methods like t-SNE may be more appropriate.

Our reply: We thank the reviewer for his/her comment. To address this, we have performed similar analyses to those conducted with PCA, but using t-distributed stochastic neighbor embedding (t-SNE). Accordingly, we have found that the t-SNE representation of the latent space of the VAE encoder displays the same behavior as observed with PCA, *i.e.*, properties with a higher value of gradient attribution map A (thus, more relevant) are qualitatively better at organizing the latent space. Selected results for PCA and t-SNE are shown in Fig. R1 and Fig. R2, respectively. The coloring of the plots corresponds to the properties in the order derived from the gradient attribution map. Indeed, the most-well organized projections are determined by properties such as exchange energy E_{XX} and kinetic energy E_{KIN} which exhibit the highest A values. Contrarily, the projections associated with properties such as the dipole moment ζ and HOMO-LUMO gap E_{GAP} , characterized by the lowest A value, are the least correlated. Moreover, we have found an overlap between the t-SNE representation of both the latent space representation from the VAE (encoding of reconstructed CMs) and property encoders (see Fig. R3), as it is obtained with PCA (see Fig. 5 of the main text). These results demonstrate that the insights about the structure of the latent space remain consistent irrespective of the dimensionality reduction technique employed. We have mentioned this conclusion in the **Interpretability and performance of our model** subsection and have added a more detailed discussion in Section 2 of the SI, see text highlighted in red.

Figure R 1 PCA representation of the latent space of the QIM model colored with different properties. The properties are ordered following the relevance criterion based on the gradient attribution map A outlined in Fig. 3 of the main text, *i.e.*, E_{XX} and ζ present the highest and lowest A values.

Figure R 2 t-SNE representation of the latent space of the QIM model colored with different properties. The properties are ordered following the relevance criterion based on the gradient attribution map A outlined in Fig. 3 of the main text, *i.e.*, E_{XX} and ζ present the highest and lowest A values..

Figure R 3 Overlap of t-SNE components between both the latent space representation from the VAE and the property encoders.

REVIEWERS' COMMENTS

Reviewer #2 (Remarks to the Author):

The authors fully addressed my concerns and the manuscript is ready to publish now.